# Aptamers Regulating the Hemostasis System

**DOI:** 10.3390/molecules27238593

**Published:** 2022-12-06

**Authors:** Anatoly A. Vaganov, Tatiana E. Taranushenko, Natalia A. Luzan, Irina A. Shchugoreva, Olga S. Kolovskaya, Polina V. Artyushenko, Tatiana N. Zamay, Anna S. Kichkailo

**Affiliations:** 1Department of Pediatrics, Faculty of Medicine, Krasnoyarsk State Medical University, 660022 Krasnoyarsk, Russia; 2Laboratory for Biomolecular and Medical Technologies, Faculty of Medicine, Krasnoyarsk State Medical University, 660022 Krasnoyarsk, Russia; 3Federal Research Center Krasnoyarsk Scientific Center, Siberian Branch of the Russian Academy of Sciences, Akademgorodok, 660036 Krasnoyarsk, Russia

**Keywords:** platelets, hemostasis, nucleic acid aptamers, thrombin, G-quadruplex, von Willebrand factor, anticoagulants, cardiovascular system

## Abstract

The hemostasis system is a complex structure that includes the fibrinolysis system, and Yes this is correct coagulation and anticoagulation parts. Due to the multicomponent nature, it becomes relevant to study the key changes in the functioning of signaling pathways, and develop new diagnostic methods and modern drugs with high selectivity. One of the ways to solve this problem is the development of molecular recognition elements capable of blocking one of the hemostasis systems and/or activating another. Aptamers can serve as ligands for targeting specific clinical needs, promising anticoagulants with minor side effects and significant biological activity. Aptamers with several clotting factors and platelet proteins are used for the treatment of thrombosis. This review is focused on the aptamers used for the correction of the hemostasis system, and their structural and functional features. G-rich nucleic acid aptamers, mostly versatile G-quadruplexes, recognize different components of the hemostasis system and are capable of correcting the functioning.

## 1. Introduction

Cardiovascular system disorders cause up to 60% of deaths among other etiological factors. Manifestations of these disorders can be both thrombotic events and hemorrhagic syndromes. Pathology of the hemostasis system is associated with high mortality and the risk of disability at any age [1,2,3,4,5,6,7]. High mortality due to thrombosis and bleeding is associated with serious gaps in the understanding of the mechanisms of the coagulation system [6,7]. Recent progress in molecular medicine led to a significant revision of the basic concepts: biomarker discovery, identification of the therapeutic molecules, and receptors for targeting.

The hemostasis system is a complex system, which includes coagulation and anticoagulant parts, and fibrinolysis. The hemostasis system consists of platelets and internal and external hemostasis pathways [7,8,9]. All components of this system are in dynamic equilibrium (Figure 1). In connection with the multicomponent nature of the system, it becomes relevant to study the key changes in the signaling pathways, and to develop new diagnostic methods and modern highly selective drugs. This may be promising in terms of making accurate diagnostics, reducing side effects, and/or the simplification of laboratory examinations.

Thrombosis of various etiologies is the most formidable and frequent manifestation of hemostasis system disorders. Classical drugs that act in this pathology are represented by direct and indirect anticoagulants and antiaggregants. The mechanism of action of the former is represented by the action on activated factors II, IX, X, and XI, with varying degrees of selectivity [10,11,12]. Modern selective drugs of this group, which have tablet forms, are not approved for use in pediatric patients. The second group of drugs act mainly on inactive factors II, V, IX, and X, reducing their amount in the bloodstream [11,13]. Both groups of drugs require frequent laboratory monitoring during the therapy. The pharmacokinetics of antiplatelet agents are focused on changing the function of platelets through the effect on the receptor apparatus of the cell (often non-selectively and/or irreversibly), by inhibiting platelet aggregation and suppressing their adhesion to the inner surface of blood vessels [14,15]. The complexity of the use of all of the above drugs is the lack of specific antidotes or a long time before the onset of a clinical effect.

Thus, the problem of regulating the activity of hemostasis components is extremely relevant. One of the ways to solve this problem is the development of molecular recognition elements capable of blocking one of the hemostasis systems and/or activating another. Such molecules can be aptamers, which can be obtained for any given target. Aptamers are single-stranded deoxyribonucleic acid (DNA) or ribonucleic acid (RNA) molecules that bind to protein targets by folding into a three-dimensional conformation, similar to antibodies [16,17]. Aptamers are most often isolated by a method called systematic evolution of ligands by exponential enrichment (SELEX), first described 25 years ago.

SELEX is an iterative process that starts with a large library of randomized nucleic acid sequences. In theory, the library contains two fixed ends to allow for reverse transcription PCR (RT-PCR) with a randomized region in the middle, typically up to 40 nucleotides. This yields a theoretical library of 4^40^ different sequences. In practice, a portion of this large library, approximately 10^15^ different molecules, is then incubated with a target protein, and the nucleic acid molecules that bind to the protein are separated. To select RNA aptamers, the bound RNAs are amplified by RT-PCR and the resulting DNA templates are subsequently transcribed. This new pool of nucleic acids is enriched for the target protein, and the process is repeated 8–12 times until an RNA pool with a high affinity for the target protein is isolated. The pool is then sequenced and characterized to identify the aptamers with the highest affinity.

Subsequent works on manipulating the structure and formulation of aptamers expanded their pharmacological properties to make them a versatile class of compounds that can be tailored to specific clinical needs.

Features of aptamers include [16,18,19]:

The entire selection is a chemical process carried out in vitro and can, therefore, target any protein;

Uniform activity regardless of batch synthesis;

Pharmacokinetic parameters can be changed on demand;

Wide variety of chemical modifications to molecules for diverse functions;

Return to original conformation after temperature insult;

Unlimited shelf-life;

No evidence of immunogenicity;

Aptamer-specific antidotes can be developed to reverse the inhibitory activity of the drug.

Many different classes of aptamers with various molecular, cellular, and tissue targets are described in the literature. Aptamers have also been designed for the treatment and prevention of thrombotic events.

The development of antithrombotic aptamers takes advantage of targets that are extracellular and uses antidote oligonucleotides that reverse aptamer activity in the setting of hemorrhages. Several coagulation factors and a platelet protein were targeted for aptamer selection and show promise to treat thrombosis. Examples of some molecules are presented in this review (Figure 1, Table 1).

The data presented in the table indicate the lower selectivity of the commonly used drugs compared to aptamers. Usually, small-molecule drugs are not targeted. Attention should be paid to the irreversibility of the action of antithrombotic medicines and, consequently, the difficulties in their clinical use, since they tend to cause bleeding. The drug influence duration is uncontrollable; therefore, the therapy should be accompanied by frequent laboratory monitoring. An important advantage of the aptamers is the antidote existence (oligonucleotide complementary to aptamer sequence). To date, aptamers have been selected for an overwhelming number of blood coagulation factors.

## 2. Aptamers to Blood Coagulation Factors

### 2.1. Aptamer to FXII/XIIa

FXII (Hageman factor) is a β-globulin that is a proenzyme from the group of proteases. Normally, FXII is inactive in the blood plasma. It is activated by collagen, kininogen, and proteolytic enzymes (kallikrein, thrombin, and trypsin). Activated FXII influences FXI and triggers the hemostasis system.

RNA aptamer R4cXII-1t selectively binds FXII. Its modification with 20-fluoropyrimidine increases its stability in plasma and saves this oligonucleotide from degradation by nucleases. Even though FXII is involved both in the hemostasis and pro-inflammatory kallikrein–kinin systems, R4cXII-1t aptamer acts particularly as an anticoagulant that inhibits FXII autoactivation, but not inflammation [20]. The mechanism of its action is associated with the fact that it interacts with the FXII/FXIIa free-chain region, which is involved in the binding of the anions and FXI [20,21,22].

### 2.2. Aptamers to FXI/XIa 

FXI has emerged as a promising target for safer anticoagulants [16,26]. FXI is a 160 kDa homodimer comprising two identical disulfide-linked polypeptide chains; specific proteolysis of the Arg369-Ile370 bond, mediated either by FXIIa or thrombin, converts FXI from an inactive precursor to enzymatically active FXIa. FXIa catalyzes the conversion of FIX to FIXa, which leads to FXa and thrombin generation. Basic and epidemiological studies indicate that FXI is important in thrombosis. 

FELIAP is an FXIa-binding DNA aptamer [25]. Anti-FXIa aptamer has a hypervariable central sequence 5′-AACCTATCGGACTATTGTTAGTGATTTTTATAGTGT-3′. K_a_ values of 5.2 ± 0.1 × 10^4^  M^−1^ S^−1^, K_d_ values of 9.5 ± 0.1 × 10^−5^  S^−1^, and K_D_ values of 1.8 ± 0.1 × 10^−9^ M were obtained (mean  ±  SD of three determinations) [26]. Firstly, FELIAP acts as a competitive inhibitor of chromogenic substrate S2366, which is a tripeptide nitroanilide compound (pyroglutamic-prolyl-arginyl-p-nitroanilide) that must, by its small size, enter the interior of the FXIa active site pocket to be cleaved and liberate the colored product nitroanilide. Secondly, it inhibits two different FXIa-dependent reactions: FXIa-mediated activation of FIX, and FXIa-mediated formation of FXIa–antithrombin complexes. While this observation does not, per se, exclude an allosteric effect of FELIAP on FXIa, it renders it less likely than active site binding. Thirdly, FELIAP does not affect FXI activation, further indicating specificity for FXIa, one of whose cardinal features is the active site. Taken together, these data suggest that FELIAP binds specifically to FXIa at or near its active site, with high affinity consistent with the observed nanomolar KD. The secondary structure of the FELIAP aptamer is represented in Figure 2a and Figure 3a [26].

Aptamers 11.16 and 12.7 are non-competitive inhibitors of the FXIa active center and activators of FIX. In normal human plasma, aptamer 12.7 significantly prolongs the clotting time. Aptamers 11.16 and 12.7 interact with FXIa ABS2 and the charged region in the FXIa autolysis loop. Aptamers activity can be halted in the event of a bleeding hazard induced by the therapy by the antidote (oligonucleotide with the sequence complimentary to the aptamers). This aptamer neutralization results in rapid reversal and controllable anticoagulation levels, and improves the safety of these drugs [24].

### 2.3. Aptamers to FX/Xa

Factor Xa (Stuart–Prower factor) plays a central role in blood clotting and is the target of several anticoagulants. FX-γ-globulin is a proenzyme (protease) that is a component of the prothrombin activator. Vitamin K is necessary for its synthesis. Under the influence of FIII, FVII, and FVIII, FIX passes into the active form Xa, which, together with FV and Ca^2+^, forms an enzyme complex that is a prothrombin activator. Factor Xa cleaves prothrombin to thrombin.

RNA11F7t aptamer, which is selective for FX, competes with Va for the binding to Xa. RNA11F7t interaction with FX suppresses the formation of an enzyme complex that activates prothrombin. Clotting time under the influence of the RNA11F7t aptamer is higher than when using the same concentrations of rivaroxaban and edoxaban [29]. RNABA4 is a bivalent made by linking RNA11F7t and R9D-14T, and works as an anticoagulant by blocking FXa and thrombin [28]. 

### 2.4. Aptamer to FIXa

FIX is responsible for prothrombinase formation. A deficiency of factor IX leads to hemorrhages.

The aptamer 9.3t blocks an extended substrate binding site to inhibit FIX-mediated activation of FX, but it does not inhibit FVIIIa/FIXa complex formation, or inhibit FIXa-mediated activation of FX without affecting FIXa/FVIIIa complex assembly [57]. The secondary structure of the 9.3t aptamer is represented in Figure 2b [31].

REG1 is the first aptamer–antidote pair to be tested in humans. Phase I clinical studies established REG1 as a safe, effective, and reproducible anticoagulation system in healthy patients or patients with stable coronary artery disease on antiplatelet therapy (aptamer/antidote system) [58]. The action of the aptamer REG2 could be stopped by an antidote system [59].

Aptamer-based anticoagulants show a good anticoagulative effect on animal models. In the work [60], the efficiency thromboprophylaxis and blood loss of the anticoagulants on the basis of the aptamer DTRI-178 and its parent aptamer 9.3t with the non-fragmented heparin was compared in pigs. Both anticoagulants reach a satisfactory and similar thromboprotective effect. Animals treated with non-fragmented heparin had increased bleeding in the area of the surgical intervention, which required more blood for transfusion compared to pigs treated with the aptamer-based anticoagulants. 

### 2.5. Aptamers to VII/VIIa

The main physiological role of FVII (proconvertin) γ-globulin is the activation of FX. Together with tissue thromboplastin, it forms a complex that activates FX. FVII is activated by tissue factor by complex formation. Thrombin additionally activates FVII [31]. Several aptamers capable of regulating the functional state of FVII have been selected. RNA aptamer 16.3 inhibits TF/FVIIa-mediated FX activation by preventing TF complexation with FVIIa [31]. The 2′fluoropyrimidine RNA aptamers 7S-1 and 7S-2 link FVIIa, exhibiting potent anticoagulant properties [32].

### 2.6. Aptamers to II/IIa (Prothrombin/Thrombin)

Thrombin is a multifunctional “trypsin-like” serine protease that plays a fundamental role in the last step of blood clotting, is able to catalyze the conversion of soluble fibrinogen into insoluble fibrin strands, while also intervening in other physiological and pathological processes [61]. Despite its pivotal function, thrombin activity needs to be repressed in some cases, e.g., coronary surgery, cancer therapy, and treatment of cardiovascular diseases, to name just a few. For instance, the inhibition of its functionality is one of the most effective antithrombotic strategies and, therefore, the development of efficient anticoagulant agents is of paramount importance [62,63,64].

Aptamers are one of the most promising anticoagulants with minor side effects and significant biological activity. Several known aptamers, which contain G-rich sequences and fold into G-quadruplex structures, have been extensively studied as targets for thrombin. G-quadruplexes are one of the most studied and biologically important non-canonical forms of DNA [65]. Recent studies have shown that G4 is formed in human genomic DNA in all phases of the cell cycle, affecting key biological processes [64]. Furthermore, G-quadruplexes constitute a structurally diverse group of compounds, which results in almost unlimited possibilities for selecting aptamers towards a great repertoire of targets [66]. All structural arrangements of G-quadruplexes possess common features, i.e., the presence of G-tetrads formed by Hoogsteen hydrogen-bonded guanines and stabilized by cations such as K^+^ or Na^+^ and others (Figure 3a). G-quadruplexes interact with different cations and the following order of ions that stabilize G-quartet is reported: Sr^2+^ > Ba^2+^ > Ca^2+^ > Mg^2+^ and K^+^ > Rb^+^ > Na^+^ > Li^+^ = Cs^+^ [67]. Typical G-quadruplexes can be divided into three main topological families: parallel quadruplexes, antiparallel quadruplexes, and hybrid quadruplexes. Quadruplexes may consist of one (monomolecular), two (bimolecular), or four (tetramolecular) separate G-rich strands (Figure 3b,c). 

#### 2.6.1. Thrombin-Binding Aptamer (TBA)

The 15 mer G-rich quadruplex, named thrombin-binding aptamer (TBA, also known as HD1) exhibits a 15 nucleotide long palindromic DNA sequence, 5′-GGTTGGTGTGGTTGG-3′, that folds into an antiparallel G-quadruplex with a chair-like topology consisting of two G-quartets [62,68]. The core of the TBA G-quadruplex is flanked by two TT and one TGT loop (Figure 2c). 

T4 and T13 form a non-canonical interloop T–T base pair. Upon folding into an antiparallel, intramolecular G-quadruplex structure with a chair-like conformation, TBA can efficiently act as a thrombin activity modulator, being able to selectively recognize the fibrinogen-binding exosite I (Figure 1) of the protein, thus, inhibiting fibrin clot formation [66]. K_D_ was about 4.5–8.2 ×  10^−9^ M [69]. Due to the high doses needed to achieve satisfactory therapeutic effects, TBA did not progress over Phase I studies in clinical trials [66]. Its high potential for therapeutic [16,70] and diagnostic [71] applications stimulated the development of novel TBA analogs designed to improve their overall properties and overcome the observed drawbacks. Following this approach, a large number of chemically modified TBA variants have been proposed in the literature [68]. These efforts have led to several optimized analogs, showing remarkably improved physicochemical and biological properties, although none of them have reached advanced clinical trial studies thus far [36].

#### 2.6.2. NU172

NU172 (ARC2172) is a nucleotide DNA aptamer selected to bind and interfere with thrombin function with the sequence 5′-CGCCTAGGTTGGGTAGGGTGGTGGCG-3′ (Figure 2d). 

This mixed duplex–quadruplex aptamer shows short-acting anticoagulation properties, as well as being well-tolerated without serious side effects [68,69]. Presumably, the presence of a flexible linker spanning both ends of NU172 might affect the architecture of the quadruplex–duplex junction, thus, also affecting the G-quadruplex loop spatial arrangement, and, in such a way, affects thrombin recognition [36]. From a structural point of view, NU172 is characterized by a duplex module combined with a minimal G-quadruplex motif, whose primary sequence differs from that of TBA not only for the additional duplex domain, which is composed of four pairs of complementary nucleotides. K_D_ was about 12  ×  10^−9^ M [68].

It binds to and inhibits exosite1 on thrombin (Figure 1). Robust in vivo studies demonstrate anticoagulation in canine cardiopulmonary bypass as well as sheep and monkey hemofiltration. The optimized version of the aptamer called NU172 is currently in human Phase IIa clinical trials [68].

NU172 aptamer was evaluated in Phase 2, an open-label clinical study in 30 patients with heart disease by ARCA biopharma. No updates are available regarding the current recruitment or completion of this study (U.S National Library of Medicine, Bethesda, MD, USA, ClinicalTrials.gov Identifier: NCT00808964) [16].

#### 2.6.3. RE31

Mazurov et al. created a 31 mer G-rich quadruplex, named RE-31, with the sequence 5′-GTGACGTAGGTTGGTGTGGTTGGGGCGTCAC-3′ (Figure 2e). It consists of both a G-quadruplex and a duplex part connected by two non-complementary nucleotide base pairs.

A study of the structure of the RE31 complex with thrombin shows that they bind through small T–T loops with the participation of the same amino acid residues from the thrombin side and by the same mechanism as TBA [68]. RE31 proves to be significantly more effective than TBA. In addition, it inhibits the prothrombotic reactions of thrombin, which suggests that the presence of the hinge and duplex region of RE31 probably increases the inhibitory activity and affinity of the aptamer for thrombin. As a drug, it is proposed to use a modified version of RE31 with protected terminal nucleotides and containing a PEG attached to the 5′ end [16]. Preclinical studies of the modified RE31 in a rat model of thrombosis were carried out, where the fundamental possibility of suppressing arterial thrombosis using this aptamer was shown [70]. Therefore, RE31 modified in this way can be a promising aptamer for the creation of a direct thrombin inhibitor for intravenous administration on its basis.

Multivalent aptamers interacting with their target proteins through multiple sites exhibit stronger effects than their monovalent counterparts. 

#### 2.6.4. ThAD 

Bivalent aptamer ThAD consists of two components binding with exosites 1 and 2. These bivalent aptamers exhibit strong allosteric attenuation of thrombin cleavage activity and exhibit an extremely potent anticoagulant effect in human plasma [72]. Shortening the aptamer increases its affinity and specificity to the target. Truncated M08 aptamer lengthens the clotting time by 10–20 times compared to anticoagulants such as TBA(HD1), NU172, RE31, and RA36 [69].

#### 2.6.5. G-Quadruplex Aptamers to Thrombin

The DNA ligand 60-18 [32] (Figure 2f) binds to the heparin-binding exosite. DNA 60-18 [32] is a quadruplex/duplex oligonucleotide. Its “core” sequence of 15 nucleotides contains eight highly conserved guanine residues involved in the formation of G-quadruplex structures. This contributes to greater stability and binding affinity for thrombin. DNA 60-18 inhibits thrombin-catalyzed fibrin clot formation in vitro [71].

Tog 25 binds both human and porcine thrombin with high affinity. Tog 25, a characteristic member, inhibits two of thrombin’s most important functions: plasma clot formation and platelet activation [34]. The full-length prothrombin aptamer R9D-14 is an 80 nucleotide long, 2′Fluoro-pyrimidine-modified RNA aptamer. R9D-14 binds prothrombin. Modified R9D-14T binds prothrombin and thrombin pro/exosite I with high affinity and inhibits both thrombin generation and thrombin exosite-I-mediated activity. R9D-14T is a more potent inhibitor than the thrombin exosite I DNA aptamer ARC-183 [35]. The secondary structure of the R9D-14T aptamer is represented in Figure 2g.

Quadruplexes with a circular structure are more stable and have higher affinity than the regular ones. CTBA4A-B1 is a circular DNA–aptamer. This aptamer demonstrates very high affinity, stability (half-life—8 h), and excellent anticoagulatory activity [36]. The secondary structure of the CTBA4A-B1 aptamer is represented in Figure 2h.

### 2.7. Aptamers to Pre-Kallikrein/Kallikrein

Kallikrein enhances FVIIa, one of the first components of the blood coagulation cascade. Pre-kallikrein, its precursor, is cleaved by FVIIa and the kininogen complex. Kall1-T4 is an RNA aptamer targeting both kallikrein (KD = 0.88 nM) and pre-kallikrein (KD = 0.28 nM). It causes a dose-dependent increase in the clotting time [37].

### 2.8. Aptamers to von Willebrand Factor 

Von Willebrand factor (VWF) is a large and complex multimeric glycoprotein essential for the initiation of hemostasis after vascular injury. VWF is the mediator of platelet adhesion to the subendothelial collagen matrix and platelet aggregation, especially at high shear rates of blood flow present in the microcirculation and stenotic arteries. Platelet adhesion involves specific sequences of the A1 domain of VWF (VWF-A1) and the platelet receptor glycoprotein Ib (GPIb) [38]. The first aptamer selected against VWF was ARC1772. This DNA aptamer binds to the GPIb domain of VWF, also known as the A-1 domain (Figure 1). 

A DNA/RNA hybrid aptamer with a 20 kD PEG moiety on the 5′ end was subsequently engineered and named ARC1779 (Figure 1, Table 1). ARC1779 showed promise in patients undergoing carotid endarterectomy by reducing thromboembolic events as measured by transcranial ultrasound, but was also associated with hemorrhage. This aptamer has a high affinity (K_D_ approximately 2 × 10^−9^ M). Furthermore, some patients had injection site reactions, echoing the concern of potential reactions to PEGylated aptamers. 

ARC1779, developed by Archemix (Cambridge, MA, USA), was investigated in ACD13-16 and TTP17-21, including limited phase II studies [39]. 

Aptamer ARC15105 inhibits platelet adhesion with the same efficiency as abciximab [41]. The stability of the aptamer was increased by its modification with four additional nucleotides. The new aptamer was named BT100. BT200, a pegylated form of the aptamer BT100, inhibits the binding of (VWF) to platelet glycoprotein GPIb (Figure 1), preventing arterial thrombosis in cynomolgus monkeys. BT200 has low nanomolar K_D_ for VWF (Figure 1, Table 1) [38,39,42]. It is being developed for secondary prevention of arterial thromboses such as stroke or myocardial infarction. Inhibition of thrombogenesis by BT200 is expected to provide a therapeutic benefit. However, there may be unexpected bleeding (e.g., incidental trauma), in which a reversal agent is required. To address this need, BT101, a complementary aptamer, was developed to specifically inhibit BT100 and BT200 functions. The research about this aptamer continues. 

Aptamer TAGX-0004 contains the artificial hydrophobic base 7-(2-thienyl)imid. Aptamer ARC15105 inhibits platelet adhesion with the same efficiency as abciximab [41]. Modification of the aptamer with four nucleotides increases the stability of the aptamer. The resulting aptamer was named BT100.azo [4,5-b]pyridine (Figure 1, Table 1) [38,42]. This aptamer targets human VWF A1 (Figure 1) with very high affinity (K_D_ = 2.2 ± 0.9 ×  10^−9^ M) and specificity [43].

To compare the effects of three agents on VWF A1, their ability to inhibit ristocetin- or botrocetin-induced platelet aggregation under static conditions was analyzed, and the inhibition of thrombus formation under high shear stress was investigated in a microchip flow chamber system. In both assays, TAGX-0004 shows stronger inhibition than ARC1779 and has comparable inhibitory effects to caplacizumab. The binding sites of TAGX-0004 and ARC1779 were analyzed with surface plasmon resonance performed using alanine scanning mutagenesis of the VWF A1 domain. An electrophoretic mobility shift assay shows that R1395 and R1399 in the A1 domain bind to both aptamers. R1287, K1362, and R1392 contribute to ARC1779 binding, and F1366 is essential for TAGX-0004 binding. Surface plasmon resonance analysis of the binding sites of caplacizumab identified five amino acids in the VWF A1 domain (K1362, R1392, R1395, R1399, and K1406).

These results suggest that TAGX-0004 possesses better pharmacological properties than caplacizumab in vitro and might be similarly promising for acquired thrombotic thrombocytopenic purpura treatment. However, there are still challenges to be overcome with this agent, such as adverse bleeding events and high costs.

Aptamers R9.3 and R9.14 bind with vWF at concentrations above 40 nM, completely inhibiting platelet plug formation. The oligonucleotide antidote AO6 completely abolishes the effect of the aptamer and reverses the antiplatelet activity [44]. DTRI-031 is another anti-vWF RNA aptamer that inhibits thrombosis in mice and platelet aggregation in whole blood. Similarly, antidote oligonucleotide quickly removes the activity of the DTRI-031 aptamer [45]. The Rn-DsDsDs-44 aptamer binds to vWF with high affinity. The stability of this aptamer is enhanced by the addition of mini-hairpin loops [46].

### 2.9. Aptamers to Activated Protein C

Activated protein C (APC) is a serine protease with anticoagulant and cytoprotective activity. The main function of APC is the regulation of the blood coagulation cascade; in particular, APC is involved in the regulation of thrombin formation. Increased production of this protein increases the risk of bleeding. Pharmacological inhibition of APC activity improves blood clotting in certain clinical situations. To suppress APC activity, several high-affinity aptamers have been obtained.

Aptamer APC-167 effectively inhibits activated protein C with a constant (Ki) of 83 nM [49]. Aptamer HS02-52G is a highly specific inhibitor of APC in plasma and whole blood. The functional activity of the aptamer is effectively blocked by the short antisense molecule AD22 [47]. The G-NB3 is a G-quadruplex aptamer for APC. It binds to the main exosite of APC (KD 0.2 nM), inhibits the inactivation of activated cofactors V and VIII, and prolongs the plasma half-life of APC, which makes the G-NB3 aptamer a promising therapeutic agent that can be used to enhance the cytoprotective functions of APC without risk of APC-related bleeding [48].

### 2.10. Aptamers to Plasminogen Activator Inhibitor 1

The plasminogen activator system plays a key role in a wide range of physiological and pathological processes. Plasminogen activator inhibitor-1 (PAI-1) is a member of the superfamily of serine protease inhibitors and the main inhibitor of tissue plasminogen activator (tPA) and urokinase-type plasminogen activator (uPA), which, in general, indicates its important role in the regulation of fibrinolysis. Fibrinolysis is the result of an interaction between several plasminogen activators and inhibitors that constitute the enzymatic cascade and ultimately leads to fibrin degradation.

Several aptamers are known to regulate PAI-1 activity and, thus, fibrinolysis. In particular, the aptamers SM-20 [51], WT-15 [52], Pionap-40 and Pionap-5 [53], and R10-4 and R10-2 [54] inhibit the anti-proteolytic activity of PAI-1.

### 2.11. Aptamer to Tissue Factor Pathway Inhibitor

The tissue factor pathway inhibitor (TFPI) negatively regulates the coagulation cascade by inhibiting the complex of factor VIIa, factor Xa, and tissue factor [25]. This group is a rare representative of aptamers, which can be used in vice versa situations, then previous molecules, such as hemorrhagic syndromes due to hemophilia. In vitro and animal studies demonstrate that the inhibition of tissue factor pathway inhibitor (TFPI) could be a potential mechanism for improving coagulation in hemophilia patients [73,74].

BAX499 (ARC19499) is a 32 nucleotide PEG-modified RNA aptamer that binds specifically to TFPI via interactions with multiple domains of the protein (Figure 1, Table 1). BAX499 inhibits the TFPI function as a negative regulator of coagulation, thereby promoting thrombin generation and improving blood clotting times. It has K_D_ =2.8 ± 0.3 × 10^−9^ M.

The phase human phase I randomized clinical trial for BAX499 was initiated in 2010 in hemophilia patients [56]. It was terminated abruptly after there was an unexpected increase in TFPI levels and an overall reduction in thrombin generation, resulting in bleeding. Scientists explained these unanticipated results after examining the inhibition of a wide range of concentrations of TFPI by BAX 499 and discovered the molecule to be a partial inhibitor of TFPI, effectively inhibiting TFPI at lower concentrations but not as well at higher concentrations. The resultant increased concentrations of TFPI predominated, leading to excessive bleeding in these patients (U.S National Library of Medicine, Bethesda, MD, USA, ClinicalTrials.gov Identifier: NCT01191372) [18].

## 3. Prospects for Selection of Aptamers to Platelet Adhesion Receptors

Despite intensive research in the field of selection of aptamers for their use in antithrombotic therapy, it should be noted that there are promising tasks in this area that have yet to be solved. One of these tasks is the selection of aptamers for one of the main platelet adhesion receptors, GP IIb/IIIa, the number of which reaches 80,000 copies on the surface of each cell. The adhesive function of GP IIb/IIIa is essential for stable platelet adhesion and aggregation. During platelet activation, signals generated within the platelet lead to conformational changes in GP IIb/IIIa from a low-affinity to a high-affinity state.

The main disadvantage of currently used GP IIb/IIIa inhibitors (tirofiban, eptifibatide, and integrillin) is the blockage of all circulating platelets, which leads to an increased risk of bleeding associated with this powerful antiplatelet strategy [75]. In addition, ligand-like inhibitors of GP IIb/IIIa used in current clinical practice, after binding to GP IIb/IIIa, can induce conformational changes, causing severe thrombocytopenia or paradoxical platelet activation [56]. Therefore, one of the promising approaches to the development of antithrombotic drugs is the specific targeting of GP IIb/IIIa, in which only activated GP IIb/IIIa is inhibited. The second promising approach is the synthesis of a safe antidote to the resulting inhibitor drug. These two tasks can be solved by selecting aptamers for this receptor.

Although existing technologies make it possible to quickly select highly specific aptamers with high affinity for a particular receptor, there are some difficulties in working with platelets associated with the peculiarities of their activation, which require the creation of specific conditions for these cells when working with them in vitro.

## 4. Future Directions

The unique characteristics of nucleic acid aptamers that represent a combination of the best features of small molecules and antibodies, including high binding affinity and specificity, non-toxic, and non-immunogenic properties, and the fact that they can be modified to have the desired properties, provides vast potential for future applications. Many in the field believe that aptamers will be particularly useful for diagnostic purposes and treatment, especially in the hemostasis system. There is a similar level of enthusiasm for their therapeutic potential in several areas, including, but not limited to, infectious disease and its complications, oncology, and inflammatory disorders. In addition, antidotes can be synthesized against any aptamer, when needed. The emergence of varying aptamers provides diversity for numerous therapeutic applications, including inhibition of active factors of coagulation, special proteins, and various receptors that facilitate prothrombotic environments at the vascular and tissue levels. Despite aptamers being promising in various applications, their clinical utility is limited by the pharmacokinetic properties of oligonucleotides. Ss DNA and RNA may degrade, be excreted, and metabolize at different rates in the body. These factors are essential for the clinical settings, and can be overcome by using different chemical modifications, which prolong the circulating half-life of the aptamers.

In the future, aptamers may be used for diagnostic procedures in children and newborns.

## Figures and Tables

**Figure 1 molecules-27-08593-f001:**
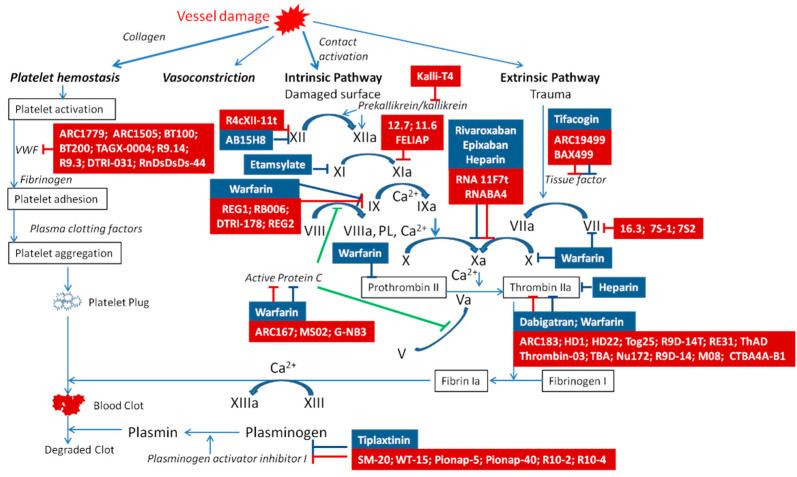
Targets for anticoagulant aptamers and drugs. The hemostasis system consists of primary (platelet) and secondary (coagulation) cascades. Coagulation hemostasis consists of intrinsic and extrinsic pathways, which are combined into a common pathway, the end product of which is insoluble fibrin. The hemostasis system is a cascade of reactions that sequentially activate coagulation factors, and its control allows regulating hemostasis. The schematic representation shows the key regulatory links, the activity of which can be inhibited by standard pharmaceuticals (shown in blue) or aptamers (shown in red). vWF—von Willebrand factor, FL—phospholipids.

**Figure 2 molecules-27-08593-f002:**
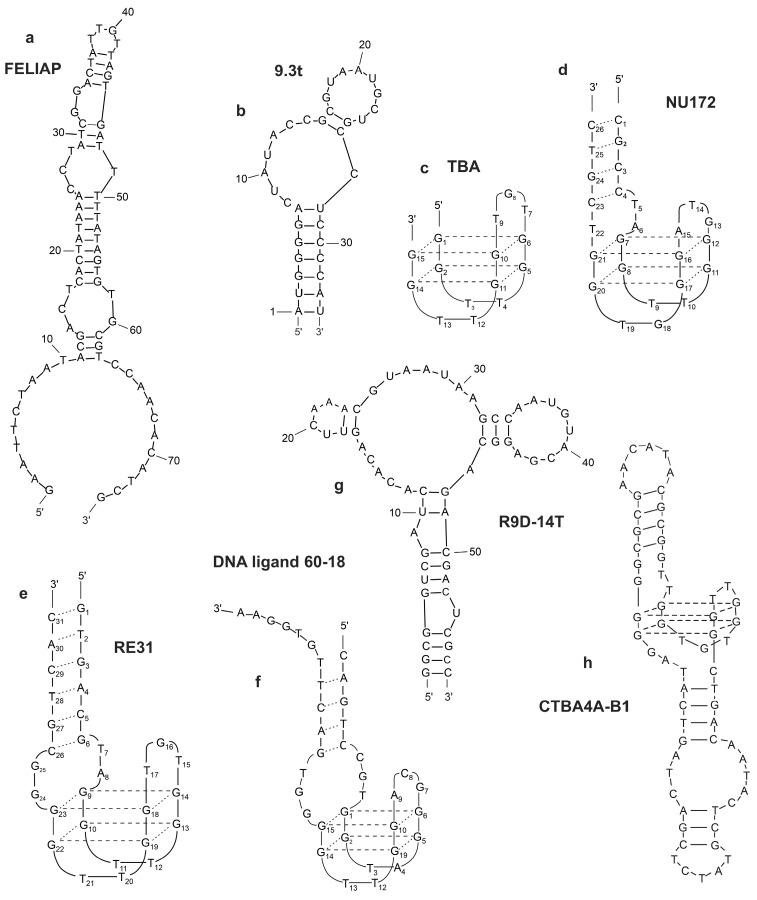
Secondary structures of G-quadruplex aptamers to thrombin: FELIAP (**a**); 9.3 (**b**), TBA (**c**); NU172 (**d**); RE31 (**e**); DNA ligand 60-18 (**f**); R9D-14T (**g**); CTBA4A-B1 (**h**).

**Figure 3 molecules-27-08593-f003:**
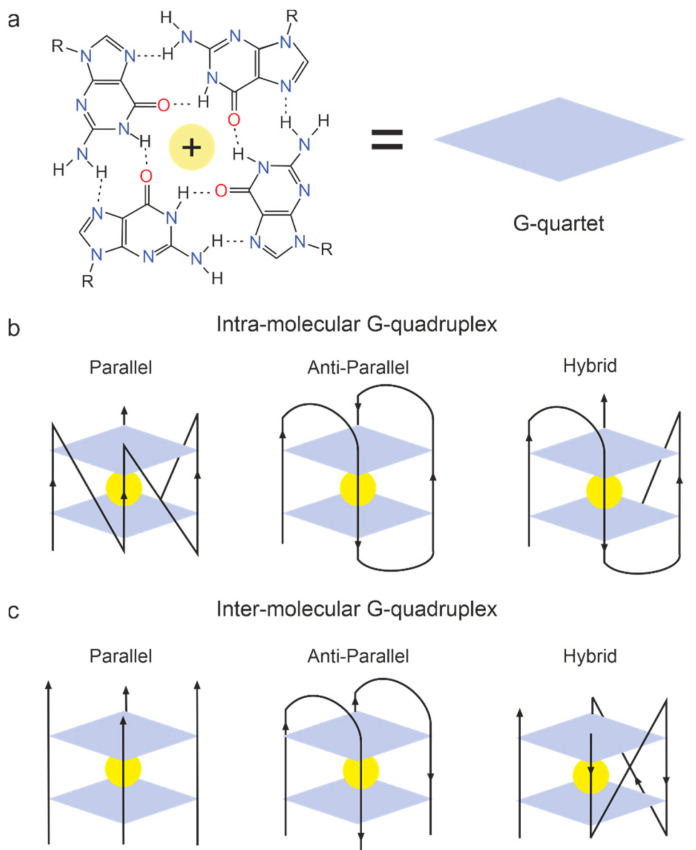
(**a**) The G-quartet structure formed by Hoogsteen base pairing of four guanine residues. (**b**) The intra-molecular topology of the G-quadruplex structure forms parallel, anti-parallel, and hybrid conformations. (**c**) The inter-molecular topology of the G-quadruplex structure forms parallel, anti-parallel, and hybrid conformations.

**Table 1 molecules-27-08593-t001:** Comparative characteristics of aptamers and some drugs that block blood coagulation factors.

Selection No.	Target	Aptamer/Antidote	Mechanism of Action/Disease/Side Effects of the Antidote	Drug/Antidote	Mechanism of Action/Disease/Side Effects of the Antidote
2.1	FXII/XIIa	R4cXII-1t/ antidote—oligonucleotide [20,21,22]	Inhibits XI cleavage by XIIa, autoactivation of XII/prevention of thromboembolic disease	Heparin/ antidote—protamine [23]	Inhibits XII/acute coronary syndrome; deep-vein thrombosis and pulmonary embolism; indwelling central and peripheral venous catheters; cerebral vascular diseases, strokes/hypotension, pulmonary edema, pulmonary vasoconstriction, pulmonary hypertension, anaphylaxis
2.2	FXI/XIa	12,7; 11,16 FELIAP/antidote—oligonucleotide [24,25,26]	Inhibits XI cleavage by XIa/prevention of thromboembolic disease	Etamsylate/ heparin/ antidote—protamine	Multifunctional hemostatic action/deep vein thrombosis, pulmonary embolism, acute coronary syndrome, percutaneous, coronary intervention, cerebral vascular diseases and strokes/hypotension, pulmonary edema, pulmonary vasoconstriction, pulmonary hypertension, anaphylaxis
2.3	FX/Xa	RNA IIF7t; RNABA4 [27,28]	Blocks FXa/FVa assembly/prevention of thromboembolic disease	Rivaroxaban/[29] heparin/ antidote—protamine	Direct inhibitor Xa/deep vein thrombosis, pulmonary embolism, acute coronary syndrome, percutaneous, coronary intervention, cerebral vascular diseases and strokes/hypotension, pulmonary edema, pulmonary vasoconstriction, pulmonary hypertension, anaphylaxis
2.4.	FIX/IXa	REG1; REG2 DTRI-178 9.3t; RB006/antidote—oligonucleotide [16,25,26,30]	Inhibits X cleavage by IXa/ acute coronary syndrome; percutaneous coronary intervention	Heparin/ antidote—protamine	Direct inhibitor FIX/deep vein thrombosis, pulmonary embolism, acute coronary syndrome, percutaneous, coronary intervention, cerebral vascular diseases and strokes/hypotension, pulmonary edema, pulmonary vasoconstriction, pulmonary hypertension, anaphylaxis
2.5.	FVII/VIIa	16,3; 7S-1; 7S-2/antidote—oligonucleotide [31,32]	Inhibits TF/FVIIa assembly/prevention of thromboembolic disease	-	-
2.6	FII/IIa	ARC183; TBA(HD1); HD22; Tog 25; R9D-14T; RE31; M08; Nu172; R9D-14; ThAD; CTBA4A-B1/antidote—oligonucleotide [33,34,35,36]	Inhibits pro/exosite Inhibits exosite II. Inhibition of fibrin formation/prevention of thromboembolic disease	Dabigatran/ heparin/ antidote—protamine	Reversible direct thrombin inhibitor/deep vein thrombosis, pulmonary embolism, acute coronary syndrome, percutaneous, coronary intervention, cerebral vascular diseases and strokes/hypotension, pulmonary edema, pulmonary vasoconstriction, pulmonary hypertension, anaphylaxis
2.7	Pre-kallikrein/kallikrein	Kalli-T4 [37]	Inhibits pre-kallikrein cleavage by IXa/prevention of thromboembolic disease	-	-
2.8	vWF	ARC1779; ARC15105; BT100; BT200; TAGX-0004; R9.14; R9.3; DTRI-031; Rn-DsDsDs-44/antidote—oligonucleotide [38,39,40,41,42,43,44,45,46]	Inhibits platelet aggregation/ thrombotic microangiopathy; thrombotic thrombocytopenic purpura; cerebral thromboembolism; cerebral vascular diseases and strokes; thrombotic thrombocytopenic purpura	-	-
2.9	Protein C	ARC167; MS02; G-NB3 [47,48,49]	Inhibits protein C/prevention of thromboembolic disease	Warfarin [50]	Inhibits protein C synthesis/deep vein thrombosis, pulmonary embolism, prevention of embolism in patients with atrial fibrillation and mechanical prosthetic heart valves, heart disease
2.10.	PAI-1	SM-20 [51], WT-15 [52], Pionap-40, Pionap-5 [53], R10-4, R10-2 [54]	Inhibits the anti-proteolytic activity of PAI-1	Tiplaxtinin/	Inhibits the anti-proteolytic activity of PAI-1 [55]
2.11	TFPI	ARC19499; BAX499 [18,25,56]	Inhibits VIIa and Xa cleavage/hemophilia	Tifacogin	Tissue factor pathway inhibitor/ severe sepsis, hemophilia

## Data Availability

The data presented in this study are contained within the article.

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
