# Peer review of "Aptamers Regulating the Hemostasis System"

_molecules, 2022, doi:10.3390/molecules27238593_

Round 1
Reviewer 1 Report
The review of Vaganov at al. presents an overview of aptamers that can target different proteins involved in the regulation of the coagulation cascade. Overall, the work is well designed and easy to read. The article is suitable as a brief minireview for those who want to get basic information on current developments in the field. This minireview is clear and informative. However, I'm not sure about the novelty, as some other similar reviews have also been published recently:
Liu, M., Zaman, K. and Fortenberry, Y.M. (2021) Overview of the Therapeutic Potential of Aptamers Targeting Coagulation Factors. International Journal of Molecular Sciences, 22.
Jin, N.Z. and Gopinath, S.C.B. (2016) Potential blood clotting factors and anticoagulants. Biomedicine & Pharmacotherapy, 84, 356-365.
Nimjee, S.M., Povsic, T.J., Sullenger, B.A. and Becker, R.C. (2016) Translation and Clinical Development of Antithrombotic Aptamers. Nucleic Acid Therapeutics, 26, 147-155.
The review is logically divided into several subsections, each dealing with aptamers directed against specific coagulation factors, which allows quick orientation in the text according to the reader's interest. The order in which the subsections are listed corresponds to the order in Table 1, which presents an overview of the aptamers depending on the type of binding site they target. To make the text even easier to navigate, I would strongly recommend using the numbering of the target categories (1. XII/XIIa, 2. XI/XIa, 3. X/Xa, 4. IX/IXa, etc.) within Table 1 to coincide with the number corresponding to each subsection (consider starting with 1 or 2, depending on whether the introduction is labelled as Section 1). To be consistent, I wonder why the information on plasminogen activator inhibitor 1 aptamers (mentioned in Section 11) is not also listed in Table 1 as in the case of other target proteins.
I have one general question:
A significant part of the article is devoted to targeting the thrombin protein and G-quadruplexes are mentioned as promising aptamers. It is well known that G-quadruplexes are highly polymorphic molecules that can adopt different topologies depending on the sequence and experimental conditions. One can then expect that G-quadruplex aptamers can be developed also for coagulants/binding sites other than those belonging to category I/IIa. Why G-quadruplex aptamers are not available/mentioned for other coagulation proteins as well?
I have a few comments and recommendations to improve the text:
1) Figure 1 should be excluded from the paper. The layout of Figure 2 is the same as Figure1 and contains all the information already presented in Figure 1. Figure 2 is essentially Figure 1 supplemented with the overview of targets for anticoagulant aptamers and drugs. Therefore, figure 2 should become figure 1 and the other figures should be renumbered accordingly. I recommend to slightly increase the size of figure 2 in the final version for better readability. Figure 1 and Figure 2 do not contain any description in the current version (and Figure 1 is not mentioned in the text at all). I believe that it would be very helpful to the reader to provide at least a brief description (in the figure caption) of the haemostasis system, targets for anticoagulation aptamers and drugs, and of less common abbreviations (with possible references to the text).
2) On page 2, line 70, it says: "This yields a theoretical library of 440 different sequences." Instead of 440, it should be 4^40 = 1.2e24.
3) The sentence starting on page 3, line 92: “Currently, according to the literature data, aptamers designed for the treatment and prevention of thrombotic events are mainly described.” I am not sure if this is true because many different classes of aptamers have been described in the literature, including those that target viruses, bacterial cells, cancer cells, etc.
4) On page 3, line 97. “ Examples of the same molecules are presented …” It's probably an accidental typo: instead of same should some.
5) On page 3, line 103. Just add the table number
6) On page 8, line 211 “ … cations such as K+ or Na+ and over (Fig 3a)“ The sentence is incomplete or something is missing after the word "over".
7) The sentence starting on page 8, line 211: Rewrite “G-quadruplexes interact with different cations and reported the following order of ions that stabilize G-quartet as …“ as “G-quadruplexes interact with different cations and the following order of ions that stabilize G-quartet is reported: …“
8) On page 10, line 292: “ … TBA, HD1, NU172, RE31, and RA36” The aptamers TBA and HD1 in this list are considered as separate entities, which is confusing. Indeed, TBA corresponds to the sequence 5′-GGTTGGTGGTTGG-3′ (known as a thrombin-binding aptamer). However, in reference 32, HD1 indicates the exact same sequence.
9) The section title (DNA 60-18) does not quite match what is inside (see page 11, line 306). Although the section contains information on the DNA aptamer 60-18 as addressed, it also mentions other aptamers Tog25, R9D-14, and CTBA4A-B1 that do not appear to have any relationship to the DNA aptamer 60-18. While tog25 and R9D-14T form hairpins, CTBA4A-B1 forms a G-quadruplex/hairpin structure.
10) The sentence starting on page 11, line 321 “CTBA4A-B1 is a circular DNA aptamer (Fig. showing very high … ” The figure number to which it refers is missing. The sequence contains mismatched parentheses: three opening and two closing, which makes difficult to understand the sentence.
The article does not contain any major shortcomings apart from the comments mentioned above, some of which are merely recommendations or typos. Despite the mentioned shortcomings, I consider the article suitable for publication in “Molecules”.
Author Response
We thank reviewers for their valuable comments and suggestions, which helped us to correct the manuscript, add the missing information.
Reviewer 2.
The review of Vaganov at al. presents an overview of aptamers that can target different proteins involved in the regulation of the coagulation cascade. Overall, the work is well designed and easy to read. The article is suitable as a brief minireview for those who want to get basic information on current developments in the field. This minireview is clear and informative. However, I'm not sure about the novelty, as some other similar reviews have also been published recently:
Liu, M., Zaman, K. and Fortenberry, Y.M. (2021) Overview of the Therapeutic Potential of Aptamers Targeting Coagulation Factors. International Journal of Molecular Sciences, 22.
Jin, N.Z. and Gopinath, S.C.B. (2016) Potential blood clotting factors and anticoagulants. Biomedicine & Pharmacotherapy, 84, 356-365.
Nimjee, S.M., Povsic, T.J., Sullenger, B.A. and Becker, R.C. (2016) Translation and Clinical Development of Antithrombotic Aptamers. Nucleic Acid Therapeutics, 26, 147-155.
Answer: Thank you very much for this observation. These reviews are indeed excellent. One of them extensively presents and characterizes aptamers to hemostasis factors (Liu M. et al., 2021). The article was cited in our review. Two other reviews published in 2016 are also undoubtedly interesting. In the review of Nimjee S.M. et al provide a summary of selected anticoagulant and antithrombotic aptamers under development. In the review by Jin, N.Z. and Gopinath potential anticoagulants based on pharmaceuticals and a single aptamer described. The purpose of this review is to compare anticoagulants commonly used to prevent hemostasis and aptamer-based anticoagulants that can be easily halted by antidotes based on complementary nucleotide sequences. In addition, in contrast to previous reviews, here we compare aptamers with the drugs, showed their role and place in hemostasis system, presented their molecular structures. All previous Reviews are cited in the text.
The review is logically divided into several subsections, each dealing with aptamers directed against specific coagulation factors, which allows quick orientation in the text according to the reader's interest. The order in which the subsections are listed corresponds to the order in Table 1, which presents an overview of the aptamers depending on the type of binding site they target.
To make the text even easier to navigate, I would strongly recommend using the numbering of the target categories (1. XII/XIIa, 2. XI/XIa, 3. X/Xa, 4. IX/IXa, etc.) within Table 1 to coincide with the number corresponding to each subsection (consider starting with 1 or 2, depending on whether the introduction is labelled as Section 1). To be consistent, I wonder why the information on plasminogen activator inhibitor 1 aptamers (mentioned in Section 11) is not also listed in Table 1 as in the case of other target proteins.
Answer: Thank you very much for the idea. The numbering in the table and sections has been changed to improve navigation. Information on plasminogen activator inhibitor 1 aptamers has been added to the table.
I have one general question
A significant part of the article is devoted to targeting the thrombin protein and G-quadruplexes are mentioned as promising aptamers. It is well known that G-quadruplexes are highly polymorphic molecules that can adopt different topologies depending on the sequence and experimental conditions. One can then expect that G-quadruplex aptamers can be developed also for coagulants/binding sites other than those belonging to category I/IIa. Why G-quadruplex aptamers are not available/mentioned for other coagulation proteins as well?
Answer: Thank you for the question. The most studied target of aptamers that suppress hemostasis is thrombin, a key component of the hemostasis system. It has been shown that, as a rule, these aptamers have a G-quadruplex structure. It can be expected that aptamers to other hemostasis factors can also have a G-quadruplex structure. However, no literature data on aptamers to other hemostasis factors were found in the literature.
I have a few comments and recommendations to improve the text:
1) Figure 1 should be excluded from the paper. The layout of Figure 2 is the same as Figure1 and contains all the information already presented in Figure 1. Figure 2 is essentially Figure 1 supplemented with the overview of targets for anticoagulant aptamers and drugs. Therefore, figure 2 should become figure 1 and the other figures should be renumbered accordingly. I recommend to slightly increase the size of figure 2 in the final version for better readability. Figure 1 and Figure 2 do not contain any description in the current version (and Figure 1 is not mentioned in the text at all). I believe that it would be very helpful to the reader to provide at least a brief description (in the figure caption) of the haemostasis system, targets for anticoagulation aptamers and drugs, and of less common abbreviations (with possible references to the text).
Answer: Fig.1 has been deleted from the manuscript. In the caption to Fig.2, a description of the components of the hemostasis system and their full names have been added.
2) On page 2, line 70, it says: "This yields a theoretical library of 440 different sequences." Instead of 440, it should be 4^40 = 1.2e24.
Answer: Thank you for the comment! This has been corrected
3) The sentence starting on page 3, line 92: “Currently, according to the literature data, aptamers designed for the treatment and prevention of thrombotic events are mainly described.” I am not sure if this is true because many different classes of aptamers have been described in the literature, including those that target viruses, bacterial cells, cancer cells, etc.
Answer: Thank you for the comment! Indeed, this sentence is not clear, we rewrite the phrase. It is marked in yellow.
4) On page 3, line 97. “ Examples of the same molecules are presented …” It's probably an accidental typo: instead of same should some.
Answer: Thank you for the comment! This has been corrected
5) On page 3, line 103. Just add the table number
Answer: Thank you for the comment! This has been corrected
6) On page 8, line 211 “ … cations such as K+ or Na+ and over (Fig 3a)“ The sentence is incomplete or something is missing after the word "over".
Answer: Thank you for the comment! This has been corrected
7) The sentence starting on page 8, line 211: Rewrite “G-quadruplexes interact with different cations and reported the following order of ions that stabilize G-quartet as …“ as “G-quadruplexes interact with different cations and the following order of ions that stabilize G-quartet is reported: …“
Answer: Thank you for the comment! We rewrite the text.
8) On page 10, line 292: “ … TBA, HD1, NU172, RE31, and RA36” The aptamers TBA and HD1 in this list are considered as separate entities, which is confusing. Indeed, TBA corresponds to the sequence 5′-GGTTGGTGGTTGG-3′ (known as a thrombin-binding aptamer). However, in reference 32, HD1 indicates the exact same sequence.
Answer: Thank you very much for your comment. The misunderstanding has been cleared. The TBA and HD1 aptamers are combined into one section it is indeed the same aptamer but with the different names.
9) The section title (DNA 60-18) does not quite match what is inside (see page 11, line 306). Although the section contains information on the DNA aptamer 60-18 as addressed, it also mentions other aptamers Tog25, R9D-14, and CTBA4A-B1 that do not appear to have any relationship to the DNA aptamer 60-18. While tog25 and R9D-14T form hairpins, CTBA4A-B1 forms a G-quadruplex/hairpin structure.
Answer: The name of the section has been changed in accordance with the material presented within the section.
10) The sentence starting on page 11, line 321 “CTBA4A-B1 is a circular DNA aptamer (Fig. showing very high … ” The figure number to which it refers is missing. The sequence contains mismatched parentheses: three opening and two closing, which makes difficult to understand the sentence.
Answer: Thank you for your comment. The text has been changed to make the information contained in it more understandable.
The article does not contain any major shortcomings apart from the comments mentioned above, some of which are merely recommendations or typos. Despite the mentioned shortcomings, I consider the article suitable for publication in “Molecules”.
Thank you very much for you time and effort to improve the manuscript.
Reviewer 2 Report
This review summarized the research on a relatively new form of molecule, aptamers, that have been generated for correction of the hemostasis system. It could serve as a useful reference for the field but the current quality is not good enough for publication. The current version lacks citation of existing work, and has too many poorly written sentences that needs to be polished.
Detailed comments:
Should cite the pioneering work of aptamer selection by the groups of Ellington and Tuerk (PMID 1697402; 2200121).
Advise to add figure legends to describe molecules marked in blue and red. I assume those marked blue are existing small molecule/antibody based drugs, whereas marked in red are aptamers.
Page 6, line 131-133, most of patients deficient in FXI does not bleed, some bleeds after major surgery at sites with high fibrinolytic activity. (PMID: 16919078)
The layout at page 7 line 173 is odd. It seems unnecessary to capitalize “Aptamers DTRI-178, 9.3t, REG1, REG2”.
Add more description of the in vivo effect of aptamers in animal models.
Should add some sentences describing some limitations of aptamers, for example they are relatively prone to be digested by nuclease in human circulation and thus some chemical modification methods have been used to prolong the circulating half life of aptamers.
The rather long sentence on G-quadruplex structure in page 8 line 201-220 is a bit odd. It is too long and i imagine this kind of structure would not be unique to thrombin aptamers. In my opinion this sentence should be shortened.
Page 12 line 340, “the GPIb domain of VWF”, should be GPIb bindng domain of VWF. Line 347-348 should merge with the preceding paragraph.
Page 13 line 423-424 “TFPI plays an integral role in negatively regulating factor VIIa in the extrinsic pathway”. This sentence seems redundant.
Page 14 line 429. “The phase human phase I randomized clinical trial”, needs correction.
Page 14 line 441, the author suggests that GP IIb/IIIa is a promising target to develop aptamers, however, no citation is given to justify the necessity/advantage of targeting this receptor over others. Previous work on this field should be cited.
Author Response
We thank reviewers for their valuable comments and suggestions, which helped us to correct the manuscript, add the missing information.
Reviewer 2.
This review summarized the research on a relatively new form of molecule, aptamers, that have been generated for correction of the hemostasis system. It could serve as a useful reference for the field but the current quality is not good enough for publication. The current version lacks citation of existing work, and has too many poorly written sentences that needs to be polished.
Detailed comments:
Should cite the pioneering work of aptamer selection by the groups of Ellington and Tuerk (PMID 1697402; 2200121).
Answer: Thank you for your comment. We cited this work as you suggested.
Advise to add figure legends to describe molecules marked in blue and red. I assume those marked blue are existing small molecule/antibody based drugs, whereas marked in red are aptamers.
Answer: Thank you very much for the comment. Explanations for the figure are provided.
Page 6, line 131-133, most of patients deficient in FXI does not bleed, some bleeds after major surgery at sites with high fibrinolytic activity. (PMID: 16919078) .
Answer: Thank you very much for the comment. The text has been changed for better understanding.
The layout at page 7 line 173 is odd. It seems unnecessary to capitalize “Aptamers DTRI-178, 9.3t, REG1, REG2”. .
Answer: Thank you very much for the comment. The structure of the section has been changed. The name Aptamers DTRI-178, 9.3t, REG1, REG2 has been removed.
Add more description of the in vivo effect of aptamers in animal models.
Answer: Data on the successful use of the aptamer DTRI-178 in pigs are presented.
Should add some sentences describing some limitations of aptamers, for example they are relatively prone to be digested by nuclease in human circulation and thus some chemical modification methods have been used to prolong the circulating half life of aptamers.
Answer: Thank you very much for the comment. This information has been added into the Disscussion section.
The rather long sentence on G-quadruplex structure in page 8 line 201-220 is a bit odd. It is too long and i imagine this kind of structure would not be unique to thrombin aptamers. In my opinion this sentence should be shortened.
Answer: Thank you very much for the comment. The text has been changed for better understanding.
Page 12 line 340, “the GPIb domain of VWF”, should be GPIb bindng domain of VWF. Line 347-348 should merge with the preceding paragraph.
Answer: Thank you very much for the comment. The line was merged with the previous paragraph.
Page 13 line 423-424 “TFPI plays an integral role in negatively regulating factor VIIa in the extrinsic pathway”. This sentence seems redundant.
Answer: Thank you very much. Unnecessary sentence has been removed.
Page 14 line 429. “The phase human phase I randomized clinical trial”, needs correction.
Answer: Thank you very much. This has been corrected.
Page 14 line 441, the author suggests that GP IIb/IIIa is a promising target to develop aptamers, however, no citation is given to justify the necessity/advantage of targeting this receptor over others. Previous work on this field should be cited.
Answer: A Reference to a paper, which describes the significance of GP IIb / IIIa and the side effects of pharmaceuticals used to block them has been inserted in the article.
Thank you very much for you time and effort to improve the manuscript.
Round 2
Reviewer 2 Report
the authors have properly addressed my comments. i have no further comments.